# Playing the Game of Universal Adversarial Perturbations

## Abstract

We study the problem of learning classifiers robust to universal adversarial perturbations. While prior work approaches this problem via robust optimization, adversarial training, or input transformation, we instead phrase it as a two-player zero-sum game. In this new formulation, both players simultaneously play the same game, where one player chooses a classifier that minimizes a classification loss whilst the other player creates an adversarial perturbation that increases the same loss when applied to every sample in the training set. By observing that performing a classification (respectively creating adversarial samples) is the best response to the other player, we propose a novel extension of a game-theoretic algorithm, namely *fictitious play*, to the domain of training robust classifiers. Finally, we empirically show the robustness and versatility of our approach in two defence scenarios where universal attacks are performed on several image classification datasets – CIFAR10, CIFAR100 and ImageNet.

## 1 Introduction

Deep learning has shown a tremendous progress in object recognition, and becomes a standard tool in many areas of computer vision such as image classification, object detection, image captioning, or visual question answering (Krizhevsky et al., 2012; Ren et al., 2015; Donahue et al., 2015; Malinowski et al., 2017). At the same time, so called adversarial samples, which can be perceptually indistinguishable from input images, can easily fool the recognition architectures by forcing them to predict wrong categories (Goodfellow et al., 2015; Moosavi-Dezfooli et al., 2016; 2017; Szegedy et al., 2014). Due to the ubiquity of deep learning, finding a robustification method is an important research direction.

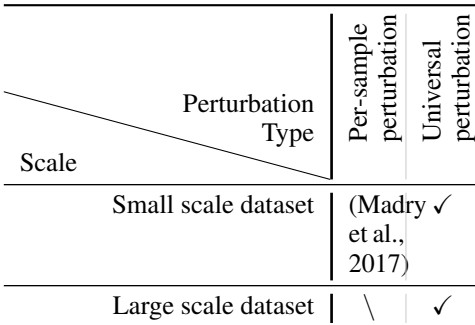

| Scale \ Perturbation Type | Per-sample perturbation | Universal perturbation |
|---|---|---|
| Small scale dataset | (Madry et al., 2017) | ✓ |
| Large scale dataset | \ | ✓ |

Table 1: Summary of defences against white box attacks. The ✓ represents the settings addressed in this paper.

In contrast to the vast body of research dedicated to per-sample adversarial perturbations (Szegedy et al., 2014; Moosavi-Dezfooli et al., 2016; Goodfellow et al., 2015), we focus on universal adversarial perturbations (Brown et al., 2017; Moosavi-Dezfooli et al., 2017). This choice is mainly motivated by the following reasons. First, those perturbations can be transferred from one datum to another, and between different architectures (Moosavi-Dezfooli et al., 2017). More specifically, they can be used to create patches that, whenever attached to real-world objects, can fool the state-of-the-art recognition architectures in the wild (Brown et al., 2017). Second, universal perturbations stand as a subclass of per-sample perturbations that is still very effective to fool deep neural networks classifiers and yet no defense strategy has addressed directly those specific perturbations. Therefore, in this work (as shown in Table 1), we only seek robustness against such universal perturbations of the inputs because we believe that they not only represent a realistic type of attacks that can easily be used widely but also because it is a sufficiently well defined problem so that an efficient defense strategy can potentially be found. We consider two categories of white-box attacks that produce universal perturbations. The first one, termed *universal perturbation*, is meant to produce a single perturbation

for the whole dataset (Moosavi-Dezfooli et al., 2017). The second one, termed *universal adversarial patch* (Brown et al., 2017), covers a part of the image with a circular patch.

To make classification more robust, we consider a novel form of adversarial training where adversarial samples are continually included in the training protocol. However, in contrast to standard adversarial training (Huang et al., 2015; Madry et al., 2017), we draw inspiration from game theory (Brown, 1951; Von Neumann and Morgenstern, 2007), and we model the adversarial training as a game between two players, let us call them *conman* and *classifier*. *Conman* fools *classifier* by generating adversarial perturbations, while at the same time, *classifier* makes robust predictions in the presence of such perturbations. In this paradigm, standard adversarial training can be seen as an approximate solution for this game, where *classifier* tries to be robust based on the most recent perturbations produced by *conman*. Instead, inspired by a vast body of the research in game theory, we propose to extend the so called *fictitious play* algorithm (Brown, 1951) that provides an optimal solution for such a simultaneous game between two players. *Fictitious play* is a more general form of adversarial training where previously generated adversarial samples are also taken into consideration.

**Contributions:** We propose a generalization of the adversarial training in the form of a simultaneous game between two players. Next, we show the framework is effective for two variants of adversarial attacks: universal perturbation and adversarial patch. In particular, it is the first defense mechanism against the adversarial patches. Compared to adversarial training, our method sacrifices less accuracy for robustness to universal perturbations, and provides better robustness to universal perturbations than the standard adversarial training. Finally, we show how our algorithm, which is inspired by *fictitious play*, scales up to large datasets such as ImageNet.

## 2 RELATED WORK

**On per-sample adversarial perturbations:** The field of finding adversarial perturbations has grown rapidly since Szegedy et al. (2014). However, this method relies on a rather expensive second-order optimization method based on L-BFGS. Later on, Goodfellow et al. (2015) have discovered that applying a single step of a sign gradient ascent (FGSM) is enough to fool a classifier. Kurakin et al. (2016) have extended the same method to use more than one iteration (I-FGSM), and hence have created a stronger, yet computationally cheap, method of attacks. Following such discoveries, more optimization methods are proposed to reliably evaluate the robustness of classifiers to adversarial samples. Whilst this area is very active, majority of methods are bench-marked and generalized in Carlini and Wagner (2017).

**On universal adversarial perturbations:** Although the field started with per-example perturbations, where each input image is separately perturbed, Moosavi-Dezfooli et al. (2017) have empirically shown the existence of so called universal perturbations. In that case, a single perturbation is applied to all the images in order to fool a classifier. This type of attack, where a single input transformation is able to fool a neural network, has found many concerning applications. For instance, it can be used to create space-bounded adversarial patches (Brown et al., 2017), or to print 3D objects that would be missclassified (Athalye et al., 2017).

**On adversarial training:** This research area aims at making classifiers more robust to adversarial attacks. To address this problem various strategies have been proposed, such as: input transformation (Liao et al., 2017; Guo et al., 2017), changing objective function (Sinha et al., 2017; Madry et al., 2017), defensive distillation (Papernot et al., 2016), or robust optimization (Madry et al., 2017). Most of these approaches have been tested against very sophisticated algorithms and very few seem to provide robustness apart from adversarial training (Uesato et al., 2017). Finally, Tramèr et al. (2017) recognize, as we do, the importance of augmenting training data with perturbations produced for more than just the current model, but this work has only been applied for black-box per-sample attacks on large datasets.

**On game theory:** *Fictitious play* is a game-theoretic algorithm meant to reach Nash equilibrium in simultaneous games (Brown, 1951). Recent work in the area of multi-agent reinforcement learning has made practical use of the *fictitious play* algorithm to learn in zero-sum or cooperative games (Heinrich et al., 2015; Heinrich and Silver, 2016; Lanctot et al., 2017; Pérolat et al., 2018). Our algorithm builds on these developments as it is an approximation of the *Fictitious play* process suited to the task of learning against universal adversarial examples.

## 3 METHOD

This section familiarizes the reader with our notations and some concepts thoroughly used in this work. Next, we detail *fictitious play* together with our novel approximation thereof. Finally, we present the various baselines used in Section 4.

### 3.1 NOTATIONS

**Classifier:** The *classifier*'s goal is to perform predictions based on the input data. This problem is addressed as a standard classification task, where the goal is to learn a function $f \in \mathcal{F}$ from data, mapping inputs to labels and generalizing well to unseen inputs, given a training dataset. This is often set up as an optimization problem, where $f$ is learned by minimizing some loss (Vapnik, 1998), that is

$$f^* \in \underset{f \in \mathcal{F}}{\operatorname{argmin}} \mathcal{L}(f, \mathcal{D}),$$

where $\mathcal{L}(f, \mathcal{D}) = \frac{1}{N} \sum_{i=1}^{N} l(f(x_i), y_i)$ is the miss-classification loss (usually the cross entropy loss) of the function $f \in \mathcal{F}$ over the training dataset $\mathcal{D} = \{(x_i, y_i)\}_{i=1}^{N}$.

**Conman:** The *conman*'s role is to fool *classifier* by producing a single perturbation that is applied to every input in the training set. More formally, for a given *classifier* $f \in \mathcal{F}$, *conman* constructs a small enough perturbation $\xi_{f,\epsilon}$ so that the perturbed dataset $\mathcal{D}_\xi$ remains 'similar' to the initial dataset $\mathcal{D}$ by solving the following optimization problem

$$\xi_{f,\epsilon} \in \underset{d(\mathcal{D}_\xi, \mathcal{D}) < \epsilon}{\operatorname{argmax}} \mathcal{L}(f, \mathcal{D}_\xi),$$

where $d(\cdot, \cdot)$ captures dissimilarity between two sets, and is defined as:

$$d(\tilde{\mathcal{D}}, \mathcal{D}) = \max_i \rho(\tilde{x}_i, x_i),$$

where $\tilde{\mathcal{D}} = \{(\tilde{x}_i, y_i)\}_{i=1}^{N}$, $\mathcal{D} = \{(x_i, y_i)\}_{i=1}^{N}$, $\rho(\cdot, \cdot)$ is a distance between samples, and $\epsilon$ is a positive real number. More precisely, we consider two kinds of perturbations. First, universal adversarial perturbations (Moosavi-Dezfooli et al., 2017) where the same perturbation $\xi$ is applied to all inputs, *i.e.* $D_\xi = \{(x_i + \xi, y_i)\}_{i=1}^{N}$ and $\|\xi\| < \epsilon$. Second, adversarial patch $\xi$, where we first sample a random location $(a, b)$ in the image, an angle of rotation $\theta$ and a rescaling factor $\chi$ that defines an affine transformation, then apply this transformation to the patch and finally overlay the transformed patch on the original image. An example of this process is shown in Figure 1. Let us call $\tilde{x}_i = A(a, b, \chi, \theta, x_i, \xi)$ the perturbed image, then the perturbed dataset is $D_\xi = \{(\tilde{x}_i, y_i)\}_{i=1}^{N}$. Learning a patch $\xi$ is done by using a gradient ascent on the cross entropy loss and a term that predicts a fixed class (see Section 3.4 for details). It is important to notice that storing a univerally-perturbed dataset $D_\xi$ comes to storing a universal perturbation (or a patch) which is cheap memory-wise.

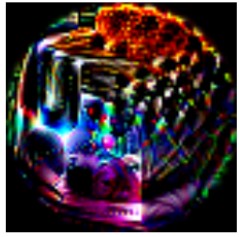
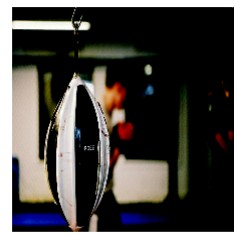
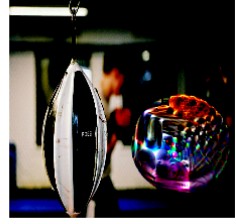

(a) Adversarial Patch        (b) Raw Image        (c) Perturbed Image

Figure 1: Example of an adversarial patch that changes the prediction of the neural network when overlaid on the raw image. Here, we trained it to predict a toaster.

**Robust classifier:** We frame the training of a robust classifier as a game between *classifier* and *conman*. This can also be expressed within an optimization framework as follows

$$f^* \in \underset{f}{\operatorname{argmin}} \max_\xi \mathcal{L}(f, \mathcal{D}_\xi).$$

Intuitively, to train a robust classifier, we need to minimize the empirical risk knowing that *conman* will attempt to maximize that risk by perturbing the dataset.

## 3.2 A Game-Theoretic Approach to Universal Attacks

Whilst popular approaches to robustification against universal adversarial attacks formulate the problem of learning a robust *classifier* as a problem of robust optimization (Madry et al., 2017), we phrase it as a game and take inspirations from the *fictitious play* algorithm (Brown, 1951). As Figures 2 and 3 illustrate, the *fictitious play* algorithm sequentially builds best responses to a uniform distribution over the past strategies of the opponent. For *conman* this means building a sequence of perturbations $\xi_n$ leading to new datasets $\mathcal{D}_n$ (Figure 2), and for the *classifier* this means to build a sequence of classifiers $f_n$ (Figure 3). In the following, we refer to the process of finding best responses as the outer-loop. At termination, the decision is taken uniformly based on the set of learned classifiers. More formally, at iteration $n \geq 1$, *conman* has to find a perturbation $\xi_n$ that maximizes the uniform average empirical loss over the past classifiers (Figure 2):

$$\mathcal{L}^n_{conman}(\mathcal{D}_\xi) = \frac{1}{n} \sum_{i=0}^{n-1} \mathcal{L}(f_i, \mathcal{D}_\xi),$$

$$\xi_n = \underset{\xi}{\operatorname{argmax}} \, \mathcal{L}^n_{conman}(\mathcal{D}_\xi), \quad \mathcal{D}_n = \mathcal{D}_{\xi_n}.$$

**Remark:** Recall that, in the case of universal perturbations, finding $D_n$ corresponds in finding a universal perturbation (or a patch) $\xi_n$ (see Section 3.4). Moreover, storing $D_n$ consists in storing $\xi_n$ (a single image) which is cheap memory-wise.

*Classifier* on its side has to find a function $f_n$ that minimizes the uniform average loss over the past datasets (Figure 3):

$$\mathcal{L}^n_{classifier}(f) = \frac{1}{n} \sum_{i=0}^{n-1} \mathcal{L}(f, \mathcal{D}_i),$$

$$f_n = \underset{f}{\operatorname{argmin}} \, \mathcal{L}^n_{classifier}(f).$$

As a convention, we write $\mathcal{L}^0_{classifier}(f) = \mathcal{L}(f, \mathcal{D}_0)$, where $\mathcal{D}_0$ is the original dataset. At termination (after $n$ steps), the strategy played $\hat{f}_n$ is a uniform random variable over the set $\{f_i\}_{i=0}^n$ of learned classifiers. Symmetrically, we define the uniform random variable $\hat{\mathcal{D}}_n$ over the set $\{\mathcal{D}_i\}_{i=0}^n$ of perturbed datasets.

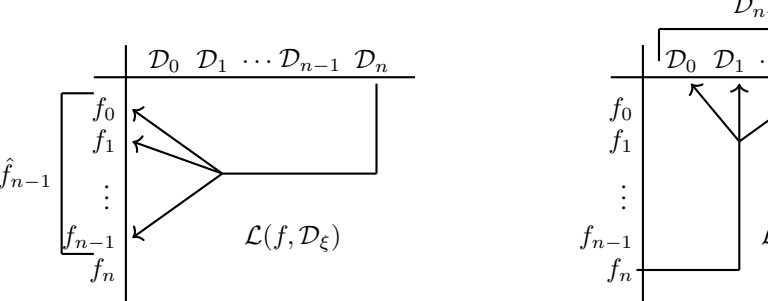

Figure 2: Illustration of the update of *conman*.     Figure 3: Illustration of the update of *classifier*.

**Approximate *Fictitious play*:** Direct application of *fictitious play* remains unpractical in our case for mainly one reason: the computation of $\hat{f}_n$ at test time requires the storage of every function $\{f_i\}_{i=0}^n$. Therefore all classifiers need to be stored in the dedicated memory (CPU or GPU memory) for rapid access to do inference. This can be a real issue for large values of $n$ as the required memory grows linearly with the number of iterations. To alleviate this problem, we leverage the fact that only $\hat{f}_n$ needs to be known to find $\xi_{n+1}$ and for the inference. Thus, instead of storing all the classifiers $\{f_i\}_{i=0}^n$, we propose to learn an approximation $\tilde{f}_n$ of $\hat{f}_n$. Observing that $\hat{f}_n$ is a uniform random

variable over the set $\{f_i\}_{i=0}^n$, where each $f_i$ minimises the loss $\mathcal{L}_{classifier}^i$, we approximate $\hat{f}_n$ by the function $\tilde{f}_n$ that minimizes:

$$\mathcal{L}(f, \{\mathcal{D}_i\}_{i \in \{0,\ldots,n-1\}}) = \frac{1}{n+1} \sum_{i=0}^{n} \mathcal{L}_{classifier}^i(f),$$

$$\tilde{f}_n = \underset{f}{\operatorname{argmin}} \, \mathcal{L}(f, \{\mathcal{D}_i\}_{i \in \{0,\ldots,n-1\}}).$$

By paying the price of this approximation, we circumvent the main drawback of *fictitious play*. Note that the second step of finding a perturbation is delegated to a separated algorithm described in Section 3.4.

---

**Algorithm 1** *Fictitious play* for adversarial perturbation (FP)

---

    **INPUT**: a dataset $\mathcal{D}_0 = \mathcal{D}$, an initial classifier $f_0$, a number of iteration $N$ and a number of steps $K$
1: **for** $n = 1, \ldots, N$ : **do**
2:     Perform $K$ steps of stochastic gradient descent on the loss $\mathcal{L}(f, \{\mathcal{D}_i\}_{i \in \{0,\ldots,n-1\}})$,
3:     Find a universal perturbation $\xi_n$ (and its corresponding dataset $\mathcal{D}_n$) that perturbs $f$. In the experiment we typically use 20000 steps of SGD (the rest of the hyper-parameters are summarized in Table 4).
4: **end for**
5: **RETURN**: $f$

---

### 3.3 RELATED APPROACHES AND BASELINES

Work on classifiers' robustness to adversarial perturbations mainly focuses on per-sample perturbations. Out of many defence approaches, the most successful is adversarial training (Madry et al., 2017) which is used as the first baseline in our experiments. Whilst the evaluation of the robustness of defence methods is still an active area of research, the robustness of this method has been assessed by two independent studies (Athalye et al., 2018; Uesato et al., 2017). It augments the batch of examples to train the network with a batch of adversarial examples $\mathcal{D}_{f,\epsilon}$. It can be seen as a stochastic gradient descent on $\frac{1}{2}\mathcal{L}(f, \mathcal{D}) + \frac{1}{2}\mathcal{L}(f, \mathcal{D}_{f,\epsilon})$ (see Algorithm 2).

Adversarial training computes per-sample perturbations of the current neural network for each batch of data to perform one gradient update. In comparison, our approach sequentially produces new perturbed data that augment the training dataset. Moreover, in our case, the perturbed datasets depend on past classifiers rather than only on the current one. As Section 3.4 shows in more details, a universal perturbation is computed in one step of the outer-loop in Algorithm 2.

The second baseline we use is the standard Stochastic Gradient Descent (SGD) on the loss $\mathcal{L}(f, \mathcal{D})$. (Algorithm 3).

---

**Algorithm 2** Adversarial Training (AT)

---

    **INPUT**: a dataset $\mathcal{D}_0 = \mathcal{D}$, an initial classifier $f_0$, a number of iteration $N$ and a number of steps $K$
1: **for** $n = 1, \ldots, N$ : **do**
2:     **for** $k = 1, \ldots, K$ : **do**
3:         Sample a batch of examples $\{(x_i, y_i)\}_{i \in B_{k,n}}$,
4:         Compute adversarial examples $\{(\tilde{x}_i, y_i)\}_{i \in B_{k,n}}$ (same procedure as in (Madry et al., 2017)),
5:         Perform gradient descent step on $\frac{1}{2} \sum_{i \in B_{k,n}} l(f(x_i), y_i) + l(f(\tilde{x}_i), y_i)$
6:     **end for**
7: **end for**
8: **RETURN**: $f$

---

---

**Algorithm 3** Stochastic Gradient Descent (SGD)

---

    **INPUT**: a dataset $\mathcal{D}$, an initial classifier $f_0$, a number of iteration $N$ and a number of steps $K$
1: **for** $n = 1, \ldots, N :$ **do**
2:     Perform $K$ steps of stochastic gradient descent on the loss $\mathcal{L}(f, \mathcal{D})$,
3: **end for**
4: **RETURN**: $f$

---

### 3.4 LEARNING UNIVERSAL ADVERSARIAL PERTURBATIONS

The state-of-the-art algorithm developed to learn universal adversarial perturbations is based on Deepfool (Moosavi-Dezfooli et al., 2017) as a subroutine and is meant to make the classifier to switch classes. This objective is different from the one that we are following in our work as we wish to find a single perturbation $\xi$ that maximizes the loss function $\mathcal{L}^n_{conman}(\mathcal{D}_\xi) = \frac{1}{n}\sum_{i=1}^{n-1} l(f(x_i + \xi), y_i)$. Thus, to find such an adversarial perturbation, we do a projected stochastic gradient ascent on the loss function with respect to the variable $\xi$. The corresponding update is:

$$\xi_{k+1} = \Gamma_\epsilon\left(\xi_k + \frac{\alpha}{|B_k|}\sum_{i \in B_k} sgn\left(\nabla_\xi l(f(x_i + \xi_k), y_i)\right)\right),$$

where $\Gamma_\epsilon$ is the projection in $\mathcal{L}_{+\infty}$-norm of size $\epsilon$ and $B_k$ is a random batch of samples of $D$.

To learn an adversarial patch, we use a similar approach. Applying a patch consists in an affine transformation of the patch and overlaying it into the image. Precisely, for a random location $(a, b)$, rotation $(\theta)$ and re-scaling $(\chi)$, we apply the canonical affine transformation defined by $(a, b, \chi, \theta)$ to the patch and overlay it on the image. This operation is differentiable. Let us note the resulting perturbed image $\tilde{x}_i = A(a, b, \chi, \theta, x, \xi)$. Therefore, we can find an adversarial patch that can fool the current network with the following update:

$$\xi_{k+1} = \mathbb{E}_{a,b,\chi,\theta}\left(\xi_k + \frac{\alpha}{|B_k|}\sum_{i \in B_k}\nabla_\xi l\left(f(A(a, b, \chi, \theta, x_i, \xi_k), y_i)\right)\right),$$

where $\mathbb{E}_{a,b,\chi,\theta}$ is the expectation over the parameters of the afine transformation applied to the patch. In practice, we handle this expectation by sampling uniformly and independently the localization over the image, the angle $\theta$, and the re-scaling parameter $(\chi)$ over a range defined in Table 4.

## 4 EXPERIMENTS AND RESULTS

This section presents our empirical findings and the metrics we use to assess them.

**Datasets considered:** CIFAR10, CIFAR100 (Krizhevsky, 2009) and ImageNet (Russakovsky et al., 2015). The last one is a large-scale dataset that is often used to benchmark recognition architectures. Moreover, models trained on this dataset are frequently believed to capture important visual features that are transferred to other computer vision problems.

**Architecture and Optimizer:** We run experiments using the same architecture (see Appendix. B) on both CIFAR10 and CIFAR100. This architecture is inspired by the VGG (Simonyan and Zisserman, 2014b) architecture without the max-pooling adjusted to work on CIFAR scaled images. However, when it comes to ImageNet, we use a Resnet 50 architecture (He et al., 2016). We use a stochastic gradient descent with momentum of 0.9 as our optimizer.

**Metric:** In all the figures, we report the accuracy of the network at the end of the iteration of the outer loop (which corresponds to 5000 steps of SGD). The *accuracy* (dotted line in the plots) is the fraction of examples that have been correctly classified for a batch of 10000 samples randomly chosen in the train, validation and test sets. The other metric we track is the *adv accuracy*. This metric represents the *accuracy* of the network under the presence of the most recent universal adversarial perturbation. Thus, the perturbation used to evaluate the performance of the network has not been used during training.

**Choice of baselines:** We compare our work to two baselines. The first one is adversarial training as this method has empirically proven it's robustness against several white box per-sample adversarial

attack and has been tested thoroughly (Uesato et al., 2017; Athalye et al., 2018). The second is standard Stochastic gradient descent as it provides a baseline when no training is done. A baseline that could have been considered would have been to adapt adversarial training by using an algorithm to generate universal adversarial perturbations. However, as generating a universal adversarial perturbation takes several order of magnitude more time than per sample perturbation, this approach is unfeasible in practice since a new perturbation needs to be generated per batch in adversarial training.

### 4.1 UNIVERSAL PERTURBATION EXPERIMENT

The first experiment is reported in Figure 4 which shows the different metrics along the learning process on the CIFAR10 dataset. This perturbation is quite large (16 pixels) compared to the ones used to train robust networks on per sample perturbation. For the classical accuracy metric, SGD performs better than the two other methods. Our *fictitious play* method suffers from a slightly worse accuracy but better than the adversarial training (Madry et al., 2017). However, for the accuracy under universal perturbation[1] our method performs better than the two others. SGD does not perform well as expected and surprisingly standard adversarial training does not provide any robustness. This can be explained by the fact that we use large perturbations (16 pixels) compared to what is usually done in the literature. Therefore, the resulting network is robust to universal perturbations only with the *fictitious play* training. On CIFAR100 (Figure 5), we run the same experiment but on smaller

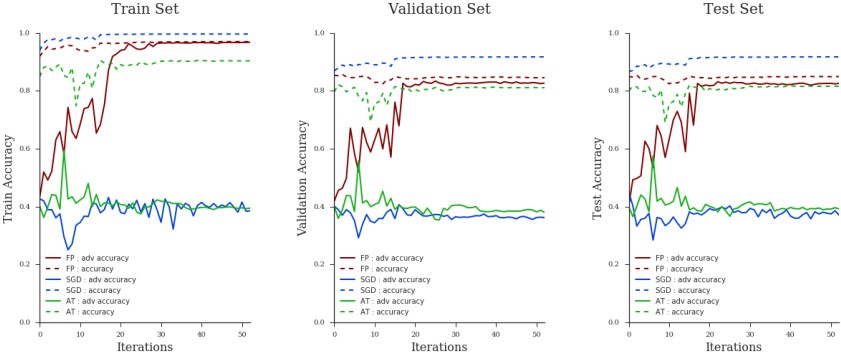

Figure 4: Results on CIFAR10 for universal adversarial perturbation. It presents the accuracy and adversarial accuracy of: Fictitious Play (FP), Stochastic Gradient Descent (SGD), and Adversarial Training (AT).

perturbations (8 pixels) instead of 16 pixels (on CIFAR10). It turns out smaller perturbations are sufficient to fool the vanilla network. In general, robustness against a class of adversarial examples seems to come at the cost of accuracy and, the larger the class of adversarial examples we try to be robust against, the larger that cost becomes. According to adversarial accuracy, our method performs well against universal perturbations (almost matching the standard accuracy). As expected the adversarial accuracy of SGD is low.

### 4.2 ADVERSARIAL PATCHES EXPERIMENT

We run a similar experiment for adversarial patches (Brown et al., 2017) and compare it with SGD as, to the best of our knowledge, there isn't any other method in the literature designed to be robust against adversarial patches. The first experiment on CIFAR100 leads to similar conclusions as the experiment on universal perturbations. The accuracy of SGD is slightly better than the one of *fictitious play* but the adversarial accuracy of our method surpasses the one of SGD. We scaled up this approach to the ImageNet dataset, and again, our approach significantly improves robustness to adversarial patches.

---

[1]Recall that we evaluate our training methods against universal perturbations whilst the majority of the work in that field focuses on per-sample perturbation.

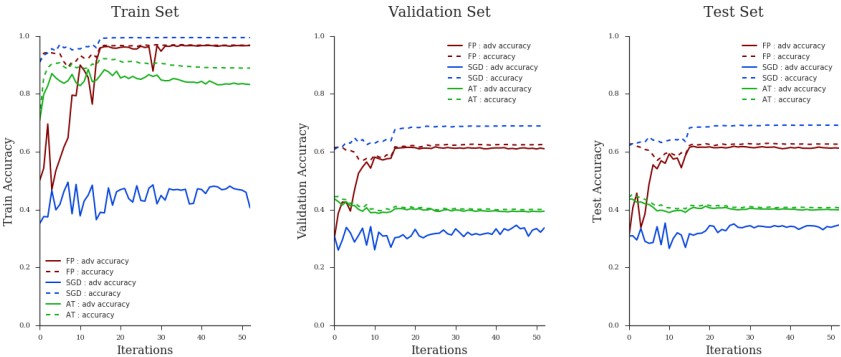

Figure 5: Results of our experiment on CIFAR100 for universal adversarial perturbation.

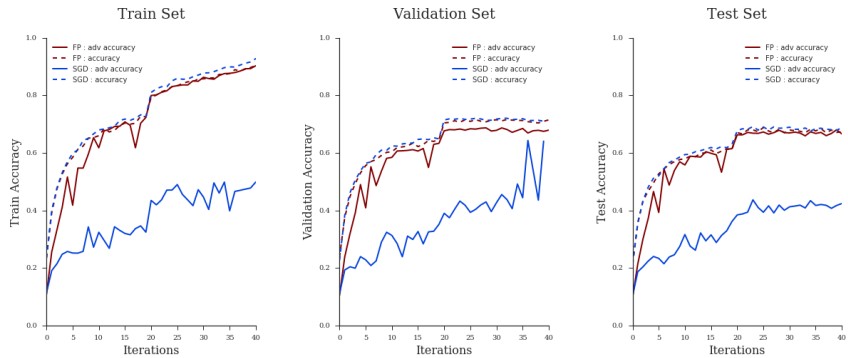

Figure 6: Results of our experiment on ImageNet for universal adversarial perturbations.

## 5 CONCLUSION

We presented a method for building classifiers robust to universal adversarial perturbations. As opposed to previous work, our method is able to learn classifiers robust to white-box attacks on large datasets whilst previous work was only able to scale on black-box attacks. Our method provides such a robustness by sequentially including universal perturbations in the training dataset. Our method surpasses standard adversarial training (Madry et al., 2017) when evaluated against universal adversarial perturbations. Moreover, our method does not only provide robustness against perturbations bounded by $\mathcal{L}_{+\infty}$-norm but also bounded in space, leading (as far as we know) to the first method that addresses robustness to adversarial patches. Many extensions of this work could be studied in other domains such as image segmentation (Metzen et al., 2017). We hope this work will help to develop a larger variety of defences against adversarial perturbations that does not only focus on per-sample perturbations but also on simpler but still realistic scenarios of universal attacks.

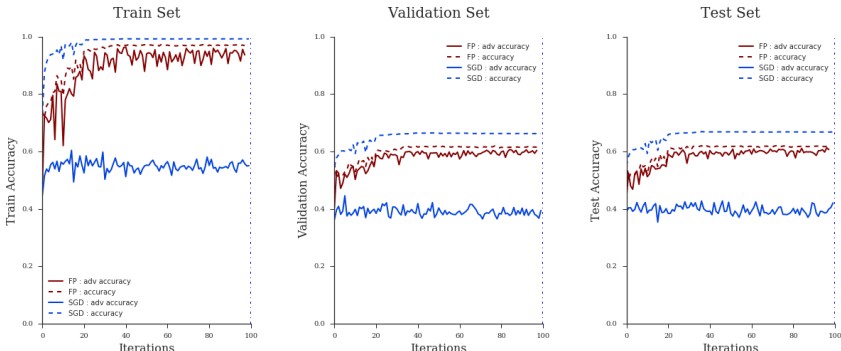

Figure 7: Results on CIFAR100 for an adversarial patch appearing at a random position on the image of a diameter 0.4 times the size of the image.

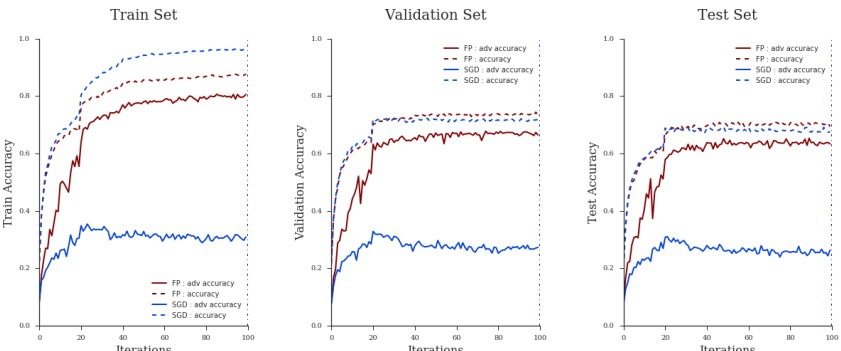

Figure 8: Results on ImageNet for an adversarial patch appearing at a random position on the image of a diameter 0.5 times the size of the image.

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

## A  HYPERPARAMETERS

In this section, we sum up the hyper-parameters used for each experiments:

| hyper-parameter Experiment \\ Scale | CIFAR10 universal perturbation | CIFAR100 universal perturbation | ImageNet universal perturbation | CIFAR100 universal perturbation | ImageNet universal perturbation |
|---|---|---|---|---|---|
| **Learning algorithm** | | | | | |
| regularization | 0.0002 | 0.0002 | 0.0002 | 0.0002 | 0.0002 |
| inner loop step $K$ | 10000 | 10000 | 5000 | 5000 | 5000 |
| start learning rate | 0.01 | 0.01 | 0.01 | 0.01 | 0.01 |
| decay $K$ | 0.1 | 0.1 | 0.1 | 0.1 | 0.1 |
| schedule $(10^4)$ | $(15, 30, 45)$ | $(15, 30, 45)$ | $(10, 20, 40)$ | $(10, 20, 40)$ | $(10, 20, 40)$ |
| Batch size | 256 | 256 | 256 | 256 | 256 |
| **Perturbations loop** | | | | | |
| initial learning rate | $210^{-5}$ | $210^{-5}$ | $510^{-4}$ | 1.0 | 1.0 |
| size universal perturbation for learning (in pixels) | 16 | 8 | 24 | / | / |
| diameter of the patch (in percentage of the image) at test $\chi$ | / | / | / | 0.4 | 0.5 |
| rotation $\theta$ | / | / | / | 20.0° | 20.0° |
| iterations number | 20000 | 20000 | 10000 | 10000 | 10000 |
| Batch size $B_K$ | 100 | 100 | 100 | 100 | 100 |

Table 2: table of hyper-parameters.

## B  NETWORK

| Architecture Used |
|---|
| Input $32 \times 32$ RGB image |
| $3 \times 3$ conv. 64 ReLU stride 1 |
| $3 \times 3$ conv. 64 ReLU stride 1 |
| $3 \times 3$ conv. 128 ReLU stride 2 |
| $3 \times 3$ conv. 128 ReLU stride 1 |
| $3 \times 3$ conv. 128 ReLU stride 1 |
| $3 \times 3$ conv. 256 ReLU stride 2 |
| $3 \times 3$ conv. 256 ReLU stride 1 |
| $3 \times 3$ conv. 256 ReLU stride 1 |
| $3 \times 3$ conv. 512 ReLU stride 2 |
| $3 \times 3$ conv. 512 ReLU stride 1 |
| $3 \times 3$ conv. 512 ReLU stride 1 |
| Fully connected layer |

Table 3: We have adapted VGG16 Simonyan and Zisserman (2014a) to the CIFAR dataset. We use batch normalization between every convolutional layer and the ReLU activation function.

# C DATASET

| Scale    size Dataset | TRAIN | TEST | VALID |
|---|---|---|---|
| Cifar 10 | 40000 | 10000 | 10000 |
| Cifar 100 | 40000 | 10000 | 10000 |
| ImageNet | 1271167 | 50000 | 10000 |

Table 4: Description of the datasets used.

