# OpenReview forum: "Playing the Game of Universal Adversarial Perturbations"
_ICLR.cc/2019/Conference_

### Official Review · AnonReviewer1 · 2018-10-30
**Nice idea, insufficient baselines**

**Rating:** 5
**Confidence:** 3

**Review:**

The authors focus solely on universal adversarial perturbations, considering both epsilon ball attacks and universal adversarial patches. They propose a modified form of adversarial training inspired by game theory, whereby the training protocol includes adversarial examples from previous updates alongside up to date attacks.

Originality: I am not familiar with all the literature in this area, but I believe this approach is novel. It seems logical and well motivated.

Quality and significance: The work was of good quality. However I felt the baselines provided in the experiments were insufficient, and I would recommend the authors improve these and resubmit to a future conference.

Clarity: The work was mostly clear.

Specific comments:
1) At the top of page 5, the authors propose an approximation to fictitious play. I did not follow why this approximation was necessary or how it differed from an stochastic estimate of the full objective. Could the authors clarify?

2) The method proposed by the authors is specifically designed to defend against universal adversarial perturbations, yet all of the baselines provided defend against conventional adversarial perturbations. Thus, I cannot tell whether the gains reported result from the inclusion of "stale" attacks in adversarial training, or simply from the restriction to universal perturbations. This is the main weakness of the paper.

3) Note that as a simple baseline, the authors could employ standard adversarial training, for which the pseudo universal pertubations are found across the current SGD minibatch.

---

> ### Author Response · Authors · 2018-11-26
> **Thanks for the review.**
>
> We would like to thank the reviewer for their comments. The reviewer considers the approach novel and well-motivated however has some concerns regarding the baselines used in the experiments. More precisely, the reviewer wants us to clarify the approximation of the fictitious play process, here are the changes done in our revised version to address their comments:
>
> 1. Approximation of fictitious play: Our method does 2 approximations of the Fictitious play process:
>     a. First as we do deep learning, we can’t guarantee that we compute a best response.
>     b. The second is that at each iteration, we do not remember all previous classifiers we computed since the beginning of the training.
>
> 2. Indeed there is a gap in the literature that we we address by proposing an algorithm specifically designed to deal with universal adversarial perturbation. However, there are no externally available baselines. Thus we choose the most natural baseline which is adversarial training.
>
> 3. This idea proposed by the reviewer would indeed interpolate between the per-sample setting and the universal perturbations setting.
> For small batches, this method would be close to per-sample perturbations and would probably be less robust at large scale (on dataset like Imagenet).
> For large batches, computing a good enough perturbation could take time. For example, with our implementation we need around 30min to get a perturbation.
>
> More specifically on the second comment which concerns the choice of the baselines. Indeed most of the methods in the literature consider conventional adversarial perturbations and here we decide to focus on universal adversarial examples. This choice is motivated by the fact that it is concretely possible to build such a universal patch and fool a recognition algorithm working in the wild. This threat can be considered even more alarming than conventional adversarial perturbations because a unique perturbation can be detrimental to an entire recognition architecture. Hence, we believe that finding a robustification algorithm in this more restrictive scenario should be addressed together or even before the conventional adversarial perturbations.  Besides, every algorithm working on conventional adversarial perturbations should adapt to the more restrictive case of universal adversarial examples. Therefore the set of baselines chosen is legitimate.

---

> > ### Comment · AnonReviewer1 · 2018-11-27
> > **thanks for the response**
> >
> > I thank the authors for a clear and helpful response.
> >
> > Thanks for clarifying the approximations to fictitious play. Id recommend also clarifying this in the text.
> >
> > I agree that the authors are addressing a valuable gap in the literature, since universal perturbations present a more realistic and limited attack scenario in production settings, as do adversarial patches as opposed to norm ball attacks.
> >
> > I understand the authors point that they provided a natural baseline from the literature. However I do still feel that their method contains two important modifications to adversarial training (limiting oneself to universal perturbations and additionally including attacks of previous models). I do think the paper would need an ablation study of these two modifications before I could recommend acceptance.
> >
> > I'd encourage the authors to provide these for a future submission.

---

### Official Review · AnonReviewer2 · 2018-10-31
**A good improvement on robust adversarial training against universal perturbations, but with questions**

**Rating:** 5
**Confidence:** 4

**Review:**

In this paper, the authors proposed universal perturbation based robust training framework. With the aid of universal perturbation, the conventional robust training framework can be further interpreted as a fictitious play. Interesting algorithm and results are reported in the paper. My detailed comments are listed as follows.

1) Some details of the proposed algorithm 1 are missing. In step 3, is just single SGD step performed? The generation of universal perturbation is not clearly discussed in Sec. 3.4. How to handle the expectation over the parameters of the affine transformation applied to the patch? MC particle-based approximation for these random parameters? If so, how many particles are used?

2) I am confused on Algorithm\,2 (AT). Is step 5 same as the robust adversarial training algorithm proposed by Madry
 et al.? What I recall is that SGD (for outer minimization) is only performed over perturbed samples, No? Please clarify it.

3) In experiments, the authors mentioned "The accuracy (dotted line in the plots) is the fraction of examples that have been correctly classified for a batch of 10000 samples randomly chosen in the train, validation and test sets." Please clearly define the train/validation/test datasets, e.g., size and how to generate adversarial examples for testing.

4) In Figure 4-6, is only the universal perturbation based attack evaluated? It does not seem a fair comparison, since the proposed min-max problem builds on the generation of universal perturbations. I wonder how robustness of the proposed method against per-sample perturbation, e.g., C\&W attack. I think it might be important to find a third-party attack method, e.g., C\&W or physically transformed attacks, to test both fictitious play and robust adversarial training.

In general, the paper contains interesting ideas and results. However, there exist questions on their implementation details and empirical results.

---

> ### Author Response · Authors · 2018-11-26
> **Thanks for the review.**
>
> We would like to thank the reviewer for their comments. The reviewer have very positive comments by highlighting that our algorithm is an interesting generalization of robust adversarial training. However, several concerns are raised concerning the clarity of the algorithms and experiments. More precisely, the first and second points of the comment section concern the algorithms. We address those comments by the following changes in our paper:
>
> Concerning step 3 of algorithm 1. In the appendix, we provided a table of hyperparameters which contains the number of steps used to compute an adversarial perturbation (see Perturbation loop). Now we add the reference of the table and give explicitly the number of SGD steps used in algorithm 1.
>
> Concerning the generation of the universal perturbation. The parameters of the affine transformation are sampled uniformly and independently for each image of the batch. The range of each parameter was given in Table 2 in the appendix. We do not use an MC particle-based approximation for these random parameters. We add that information in the main text in Sec 3.4 to make it clearer.
>
> In the robust adversarial training algorithm proposed by Madry, the outer minimization is performed only on perturbed samples. Here, we do a mix (50/50) between perturbed and non-perturbed samples. We observe that it gives good results in practice.
>
> Then points 3 and 4 raise some concerns about the experiments. We address those comments by the following changes in our paper:
> Concerning the number of samples for each datasets (We add those numbers in the appendix):
> Cifar10: Train set: 40000, Test set:10000, Valid set:10000
> Cifar100: Train set:40000, Test set:10000, Valid set:10000
> ImageNet: Train set: 1271167, Test set: 50000, Valid set: 10000.
> Concerning the robustness against per sample perturbation: We do not expect our model to be robust to per-sample perturbation and never claimed such a robustness. Providing such a robustness is quite difficult in Cifar10 and is still out of reach in large dataset such as Imagenet. Nonetheless, we do think that providing robustness to universal perturbation is still a useful technique as these perturbations can be deployed in the real world (https://arxiv.org/pdf/1712.09665.pdf).

---

> > ### Comment · AnonReviewer2 · 2018-12-02
> > **Thanks for the response & further questions**
> >
> > Thanks for the detailed response for my previous questions.
> >
> > "We add that information in the main text in Sec 3.4 to make it clearer."
> >
> > The last paragraph of Sec. 3.4 is still unclear.
> >
> > First the authors mentioned "where Ea,b,χ,θ is the expectation over the parameters of the affine transformation applied to the patch. In practice, we handle this expectation by sampling uniformly and independently the localization over the image, the angle θ, and the re-scaling parameter (χ) over a range defined in Table 4."
> >
> > Question 1: How many i.i.d. samples you drew for each parameter, a, b, \xi, \theta? I supposed that given finite-number parameter realizations, the expectation will be written in its empirical sum form, right? Do you need sampling at each iteration to update the universal perturbation?
> >
> > Question 2: Table 4 or Table 2?
> >
> >
> > " Nonetheless, we do think that providing robustness to universal perturbation is still a useful technique as these perturbations can be deployed in the real world (https://arxiv.org/pdf/1712.09665.pdf)."
> >
> > This motivation is not enough, since generating a per-sample perturbation is much easier than the universal perturbation, right? If so, we need a better motivation on the usefulness of this technique.

---

### Official Review · AnonReviewer3 · 2018-11-02
**nicely presented ideas, lacking discussion around guarantees (or not)**

**Rating:** 6
**Confidence:** 1

**Review:**

Being familiar but not an expert in either game theory or adversarial training, my review will focus on the overall soundness of the proposed method

Summary:

The authors propose to tackle the problem of adversarial training.
Deep networks are know to be susceptible to adversarial attacks.
Adversarial training is concerned with the training of networks that both achieve good performance for the original task while being robust to adversarial attacks.

They propose to focus on universal adversarial perturbations, as opposed to per-sample perturbations. The latter is a subclass of the former.
It doesn’t strike as the most natural scenario: I can’t really think of a practical image classification scenario where one would want to perturb a whole dataset of image with a single perturbation. That said, this focus leads to simpler algorithms (complexity and storage wise) which are worth exploring.

The authors first present the min-max problem of adversarial training at hand where a classifier f mimizes a loss L for a dataset D, while the conman maximizes the loss over perturbation of the dataset \epsilon.
They then introduce an algorithm to solve it inspired by fictitious play:
A sequence of classifiers and perturbed datasets are created iteratively by the two players (classifier, conman) and each player uses the complete history of its opponent to make its next move.

The objective solved by each player  is :
conman: fool all past classifiers with a single new perturbation
classifier: be robust to all past perturbations so far.

Although it makes intuitive sense, it is unclear from the manuscript whether this formulation provides any convergence guarantees. It would be great to know whether the connection to fictitious play is purely inspirational or if any of the theoretical guarantees from game theory apply here.

The conman’s objective to fool all past classifiers is the bottleneck (in terms of storage) and an approximation is proposed: the mean loss over past classifiers is replaced by the loss under a single ‘average’ classifier trained on all past dataset, with the intuition that this average classifier summarizes all past classifiers

A particular algorithm for perturbation learning is described and the proposed algorithm is compared against two baselines: a pre-existing adversarial training algorithm, an non-adversarial algorithm

The metrics chosen are accuracy and adversarial accuracy.
On standard classification tasks, adversarial algorithms perform slightly less well on the original task (accuracy) but are robust to perturbation as expected,

It would be interesting to know if these good performances extend to per-sample perturbations: Do a network trained on universal perturbations perform well against per sample perturbation?


Remarks:
sgn missing in the adversarial patch update (and who is alpha?)
introduce terminology: white box black box

---

> ### Author Response · Authors · 2018-11-26
> **Thanks for the review.**
>
> Thanks a lot for your comments on the paper. The reviewer acknowledged that the idea is novel and asks clarifications on the scenario of the attack and on the approximation made.
> Concerning the scenario: Many work on the use of universal perturbations to fool classifier have been developed. For instance, the work on universal patches (https://arxiv.org/pdf/1712.09665.pdf) can be used in the real world. In this paper the authors use a printed patch to fool an image recognition application (the Demitasse application). In our paper, we propose a method to address that problem.
> Concerning the approximation made: As we are dealing with deep neural networks, we are making two approximation to the Fictitious play process. The first one is that at each steps we do not compute the exact best response. The second is that we don’t actually maintain all the best responses computed in the past to compute a perturbation against a uniform mixture of these best response but rather learn this uniform mixture.
> We do not expect this method to give any form of robustness to per-sample perturbations. This is not the claim we are trying to support in this paper.

---

### Public Comment · (anonymous) · 2018-09-30
**The title attracted me**

But the content is not. The most critical question of generating adversarial samples is how to keep the concept class unchanged. If we are able to do so, the training is straightforward.

---

### Public Comment · (anonymous) · 2018-10-02
**Game theory perspective is not novel**

The abstract makes it sound as if it is an original contribution of this paper to view adversarial examples from the perspective of game theory:
"While prior work approaches this problem via robust optimization, adversarial training, or input transformation, we instead phrase it as a two-player zero-sum game. In this new formulation, both players simultaneously play the same game, where one player chooses a classifier that minimizes a classification loss whilst the other player creates an adversarial perturbation that increases the same loss when applied to every sample in the training set."

Here are several prior works that explicitly describe adversarial examples in terms of a minimax game:
https://pdfs.semanticscholar.org/b5ec/486044c6218dd41b17d8bba502b32a12b91a.pdf
https://ieeexplore.ieee.org/document/7966196
https://arxiv.org/pdf/1803.01442.pdf
http://openaccess.thecvf.com/content_ICCV_2017/papers/Oh_Adversarial_Image_Perturbation_ICCV_2017_paper.pdf
Including some works that mention fictitious play:
https://aaai.org/ocs/index.php/FSS/FSS17/paper/download/15994/15311
https://arxiv.org/pdf/1809.06784.pdf

There are several more such works, easily discovered by a search engine.

I haven't read the paper at all and it's possible that a related work section acknowledges this prior work exists / it's possible that this paper has a new take on games / fictitious play. Even if that is the case, the abstract really should state the novelty differently.

---

### Public Comment · (anonymous) · 2018-10-02
**Universal adversarial perturbations were known since 2014**

The introduction says "In contrast to the vast body of research dedicated to per-sample adversarial perturbations (Szegedy et al., 2014; Moosavi-Dezfooli et al., 2016; Goodfellow et al., 2015), we focus on universal adversarial perturbations (Brown et al., 2017; Moosavi-Dezfooli et al., 2017)". Actually the Goodfellow et al 2015 paper describes how to make perturbations that are independent of the input, by adding a weight vector of a linear classifier to the input. Though the examples are made using a linear classifier, they also transfer to deep nets. Andrej Karpathy later demonstrated more of these input-independent examples: http://karpathy.github.io/2015/03/30/breaking-convnets/ The catchy name "universal adversarial perturbations" wasn't coined until later but the idea already existed.

I just intend this as feedback for revision, I don't intend for this to have any bearing on whether the paper is accepted or rejected.

---

### Meta-Review · Area_Chair1 · 2018-12-14

**Confidence:** 4
**Recommendation:** Reject

**Metareview:**

Reviewers mostly recommended to reject after engaging with the authors, with one reviewer slightly suggesting to accept, but with confidence 1. Please take reviewers' comments into consideration to improve your submission should you decide to resubmit.